# Genome editing enables reverse genetics of multicellular development in the choanoflagellate *Salpingoeca rosetta*

**David S Booth[†]\*, Nicole King**

Howard Hughes Medical Institute and Department of Molecular and Cell Biology, University of California, Berkeley, Berkeley, United States

**Abstract** In a previous study, we established a forward genetic screen to identify genes required for multicellular development in the choanoflagellate, *Salpingoeca rosetta* (Levin et al., 2014). Yet, the paucity of reverse genetic tools for choanoflagellates has hampered direct tests of gene function and impeded the establishment of choanoflagellates as a model for reconstructing the origin of their closest living relatives, the animals. Here we establish CRISPR/Cas9-mediated genome editing in *S. rosetta* by engineering a selectable marker to enrich for edited cells. We then use genome editing to disrupt the coding sequence of a *S. rosetta* C-type lectin gene, *rosetteless*, and thereby demonstrate its necessity for multicellular rosette development. This work advances *S. rosetta* as a model system in which to investigate how genes identified from genetic screens and genomic surveys function in choanoflagellates and evolved as critical regulators of animal biology.

**\*For correspondence:**
David.Booth@ucsf.edu

**Present address:** [†]Department of Biochemistry and Biophysics, University of California, San Francisco, San Francisco, United States

**Competing interests:** The authors declare that no competing interests exist.

## Introduction

As the sister group of animals (*Figure 1A*), choanoflagellates have great potential for revealing the origins of animal development and the cell biology of multicellularity (*Lang et al., 2002*; *Burger et al., 2003*; *Carr et al., 2008*; *Ruiz-Trillo et al., 2008*; *Grau-Bové et al., 2017*). Comparative genomic studies have demonstrated that choanoflagellates express genes that are necessary for animal development (*King et al., 2008*; *Fairclough et al., 2013*; *Richter et al., 2018*; *López-Escardó et al., 2019*), including genes for intercellular adhesion (e.g. cadherins: *Abedin and King, 2008*; *Nichols et al., 2012*), signaling (e.g. receptor tyrosine kinases and CamKII: *Manning et al., 2008*; *Pincus et al., 2008*; *Bhattacharyya et al., 2016*; *Amacher et al., 2018*), and cellular differentiation (e.g. myc, STAT, and p53: *Young et al., 2011*; *de Mendoza et al., 2013*). Moreover, choanoflagellates and animals are the only clades that have cells with a collar complex (*Leadbeater, 2015*; *Brunet and King, 2017*), a unique cellular module in which a collar (*choano* in Greek) of actin-filled microvilli surrounds an apical flagellum (*Figure 1B*; *Sebé-Pedrós et al., 2013*; *Peña et al., 2016*; *Colgren and Nichols, 2020*). Together, these observations have motivated the development of choanoflagellates as models for researching the function and evolution of core developmental regulators (*King, 2004*; *Hoffmeyer and Burkhardt, 2016*; *Sebé-Pedrós et al., 2017*; *Brunet and King, 2017*).

The choanoflagellate *Salpingocea rosetta* has received the greatest investment in tool development (*Hoffmeyer and Burkhardt, 2016*). Its 55.44 megabase genome encodes ~11,629 genes, some of which are homologs of integral regulators for animal development (*Fairclough et al., 2013*). Moreover, the life history of *S. rosetta* provides a rich biological context for investigating the functions of intriguing genes (*King et al., 2003*; *Fairclough et al., 2010*; *Dayel et al., 2011*; *Levin and King, 2013*; *Woznica et al., 2017*). For example, *S. rosetta* develops into multicellular, spheroidal colonies called rosettes through serial cell divisions from a single founding cell (*Fairclough et al., 2010*; *Laundon et al., 2019*; *Larson et al., 2020*), a process induced by

environmental bacteria that can also serve as a food source (*Figure 1C*; *Alegado et al., 2012*; *Woznica et al., 2016*). Thus, rosette development can provide a phylogenetically relevant model for discovering genes that mediate multicellular development and bacterial recognition in choanoflagellates and animals.

A forward genetic screen was established to hunt for mutants that were unable to develop into rosettes and resulted in the identification of genes required for rosette development (*Levin et al., 2014*; *Wetzel et al., 2018*). The first of these (*Levin et al., 2014*), *rosetteless* encodes a C-type lectin protein that localizes to the interior of rosettes (*Figure 1D–E*). As C-type lectins are important for mediating intercellular adhesion in animals (*Drickamer and Fadden, 2002*; *Cummings and McEver, 2015*), this discovery highlighted the conserved role of an adhesion protein family for animal and choanoflagellate development. However, the screen also underscored the necessity for targeted genetics in *S. rosetta*. Because of inefficient mutagenesis in *S. rosetta*, forward genetics has been laborious: out of 37,269 clones screened, only 16 rosette-defect mutants were isolated and only three of these have been mapped to genes (*Levin et al., 2014*; *Wetzel et al., 2018*). Establishing genome editing would accelerate direct testing of gene candidates identified through forward genetic screens, differential gene expression, and/or genomic comparisons.

Therefore, for the present study, we sought to establish CRISPR/Cas9 genome editing in *S. rosetta*. Cas9-mediated genome editing (*Jinek et al., 2012*; *Jinek et al., 2013*; *Cong et al., 2013*) has been crucial for advancing genetics in emerging models (*Gilles and Averof, 2014*; *Harrison et al., 2014*; *Momose and Concordet, 2016*). Depending on the DNA repair pathways expressed in a given organism (*Yeh et al., 2019*), the delivery of the Cas9 endonuclease bound to a programmable guide RNA (gRNA) can direct DNA cleavage at a target site to introduce mutations from co-delivered DNA templates or from untemplated repair errors that cause insertions or deletions (*Rouet et al., 1994*; *Choulika et al., 1995*; *Bibikova et al., 2001*; *Jinek et al., 2013*; *Cong et al., 2013*). While the delivery of macromolecules into choanoflagellate cells has been a longstanding barrier for establishing reverse genetic tools, we recently established a robust method to transfect *S. rosetta* with DNA plasmids for expressing transgenes (*Booth et al., 2018*), which allowed us to perform genetic complementation (*Wetzel et al., 2018*). Despite having established a method for gene delivery in *S. rosetta*, the lack of knowledge about DNA repair mechanisms in choanoflagellates and low-transfection efficiency (~1%) presented challenges for establishing genome editing, particularly without a proven selectable marker in an endogenous gene to enrich for editing events.

Here we report a reliable method for genome editing to perform reverse genetics in *S. rosetta* that we have developed into a publicly-accessible protocol (https://dx.doi.org/10.17504/protocols.io.89fhz3n). First, we engineered a selectable marker for cycloheximide resistance as an initial demonstration of genome editing with CRISPR/Cas9 (*Figure 2*). We then inserted a foreign sequence into *rosetteless* that eliminates its function, confirming the importance of this gene for multicellular rosette development (*Figure 3*). Finally, we found that, even in the absence of selection, *S. rosetta* preferentially uses DNA templates to introduce mutations for genome editing (*Figure 4*). This work establishes genome editing in *S. rosetta* and provides a path for testing the function of choanoflagellate genes that are implicated in the early evolution of animals.

## Results

### A marker to select for cycloheximide resistance facilitates genome editing in *S. rosetta*

Our initial attempts to target rosetteless for genome editing in *S. rosetta* were either unsuccessful or resulted in editing events that were below the limits of detection. Therefore, suspecting that genome editing in *S. rosetta* might prove to be challenging to establish, we first aimed to introduce a mutation in an endogenous gene that would confer antibiotic resistance and allow selection for rare genome editing events.

In *Chlamydomonas* (*Stevens et al., 2001*) and Fungi (*Kawai et al., 1992*; *Dehoux et al., 1993*; *Kondo et al., 1995*; *Kim et al., 1998*), specific mutations in the ribosomal protein gene *rpl36a*

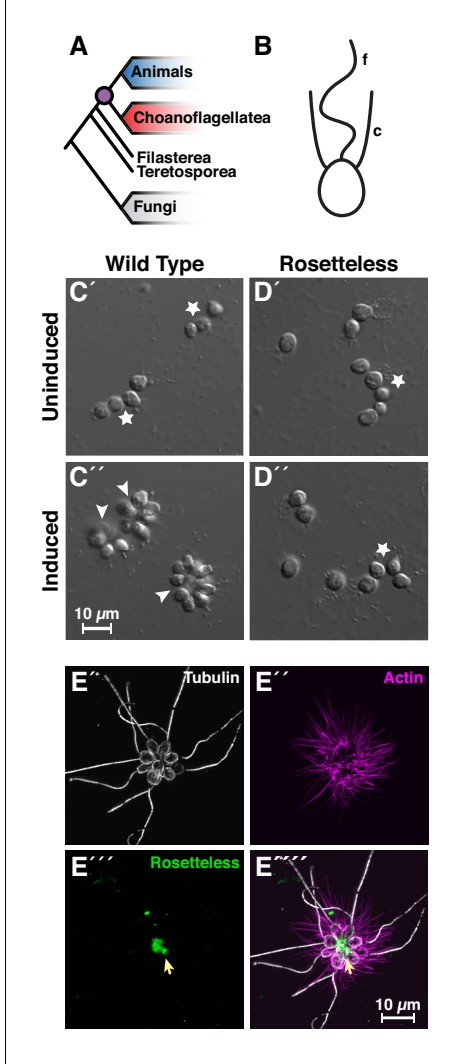

**Figure 1.** Introduction to *Salpingoeca rosetta* as a simple model for multicellularity and the ancestry of animal cell biology. (A) Choanoflagellates (blue) are the closest living relatives of animals (red) and last shared a common ancestor (purple) ~800 million years ago (*Parfrey et al., 2011*). (B) The collar complex, an apical flagellum (f) surrounded by a collar (c) of actin-filled microvilli, typifies choanoflagellates and is uniquely shared between choanoflagellates and animals (*Brunet and King, 2017*). (C) Wild-type *S. rosetta* forms multicellular rosette colonies in response to rosette inducing factors (RIFs) secreted by environmental bacteria. In the absence of RIFs (C′), *S. rosetta* grows as single cells or as a linear chain of cells (star). Upon the addition of RIFs (C′; *Alegado et al., 2012*; *Woznica et al., 2016*), *S. rosetta* develops into spheroidal, multicellular rosettes (arrowhead) through serial cell divisions (*Fairclough et al., 2010*). (D) The *rosetteless* C-type lectin gene is necessary for rosette development. A mutation in *rosetteless* allows normal cell growth as single cells and linear chains in the absence of RIFs (D′) but prevents rosette development

*Figure 1 continued on next page*

confer resistance to the antibiotic cycloheximide by disrupting cycloheximide binding to the large subunit of eukaryotic ribosomes (*Stöcklein and Piepersberg, 1980*; *Schneider-Poetsch et al., 2010*; *Garreau de Loubresse et al., 2014*). After finding that *S. rosetta* cell proliferation was inhibited by cycloheximide (*Figure 2—figure supplement 1A*), we sought to establish a cycloheximide-resistant strain through genome editing. By combining prior genetic findings (*Bae et al., 2018*) with our own structural modeling (*Figure 2—figure supplement 1B*) and bioinformatic analyses (*Figure 2—figure supplement 1C*) of the *S. rosetta rpl36a* homolog (PTSG_02763), we predicted that converting the 56$^{th}$ codon of *rpl36a* from a proline to a glutamine codon (*rpl36a$^{P56Q}$*) would render *S. rosetta* resistant to cycloheximide (*Figure 2—figure supplement 1D*). Insertion or deletion mutations that could arise as errors from repairing the double-stranded break without a template would likely kill cells by disrupting the essential function of *rpl36a* for protein synthesis (*Bae et al., 2018*).

To edit the *rpl36a* gene in *S. rosetta*, we first designed a gRNA with a 20 nt sequence from *rpl36a* to direct Cas9 from *Streptomyces pyogenes* (*Sp*Cas9) to cut at *S. rosetta* supercontig 6: 948,122 nt (*Fairclough et al., 2013*). Then we made a DNA repair template as a single-stranded DNA oligonucleotide with a sequence encoding the Pro56Gln mutation and 200 bases of flanking homologous sequence from *rpl36a* centered on the cleavage site (*Figure 2—figure supplement 2A*). To deliver the *Sp*Cas9/gRNA ribonucleoprotein complex (*Sp*Cas9 RNP) and the repair template encoding the Pro56Gln mutation into *S. rosetta* cells, we used a nucleofection protocol adapted from our recently developed method for transfecting *S. rosetta* (*Figure 2A*; *Booth et al., 2018*). We favored delivering the *Sp*Cas9 RNP rather than expressing *Sp*Cas9 and gRNAs from plasmids, as RNA polymerase III promoters for driving gRNA expression have not yet been characterized in *S. rosetta* and the overexpression of *Sp*Cas9 from plasmids can be cytotoxic for other organisms (*Jacobs et al., 2014*; *Jiang et al., 2014*; *Shin et al., 2016*; *Foster et al., 2018*) as well as increase the likelihood of introducing off-target mutations (*Kim et al., 2014b*; *Liang et al., 2015*; *Han et al., 2020*). After growing transfected cells in the presence of cycloheximide for five days, Sanger sequencing of PCR-amplified *rpl36a* showed that *rpl36a$^{P56Q}$* was the major allele in the population (*Figure 2—figure supplement 2*, compare B and C). Sequencing a clonal strain established from this population confirmed

*Figure 1 continued*
in the presence of RIFs (D'; *Levin et al., 2014*). (E) Wild-type *S. rosetta* secretes Rosetteless protein from the basal ends of cells into the interior of rosettes. Shown is a representative rosette stained with an antibody to alpha-tubulin to mark cortical microtubules and the apical flagellum of each cell (E', grey) phalloidin to mark actin-filled microvilli (E', magenta), and an antibody to Rosetteless protein (E''', green). A merge of alpha-tubulin, phalloidin, and Rosetteless staining shows that Rosetteless protein localizes to the interior of rosettes (arrow) where cells meet at their basal ends (E''''; *Levin et al., 2014*).

the *rpl36a*$^{P56Q}$ genotype (*Figure 2—figure supplement 2D*), and growth assays showed that the *rpl36a*$^{P56Q}$ strain proliferated better than the wild-type strain in increasing concentrations of cycloheximide (*Figure 2B*; Two-Factor ANOVA: $p < 10^{-20}$).

The ability to engineer cycloheximide resistance additionally offered a simple assay to optimize essential parameters for genome editing in *S. rosetta*. Therefore, we tested how varying delivery conditions would impact the frequency of template-mediated mutagenesis and, ultimately, the cell density and consensus genotype of a cell population after genome editing and cycloheximide treatment (*Figure 2—figure supplement 2E–H*). Through this optimization process (*Figure 2—figure supplement 2*), we found that efficient genome editing required transfection with at least 20 pmol of *Sp*Cas9 RNP and more than 200 nmol of a single-stranded DNA repair template that had 50 bases of homology flanking a designed mutation. Henceforth, these parameters established baseline conditions for designing and executing genome editing experiments.

## Targeted disruption of rosetteless demonstrates its essentiality for multicellular rosette development

We next sought to use genome editing as a general tool for reverse genetics in choanoflagellates. To this end, we targeted *rosetteless* (*rtls*), one of only three genes known to be required for rosette development in *S. rosetta* (*Levin et al., 2014*; *Wetzel et al., 2018*). A prior forward genetic screen linked the first rosette defect mutant to an allele, *rtls*$^{tl1}$, in which a T to C transition in the 5'-splice site of intron 7 (*S. rosetta* supercontig 8: 427,804 nt; *Fairclough et al., 2013*) was associated with

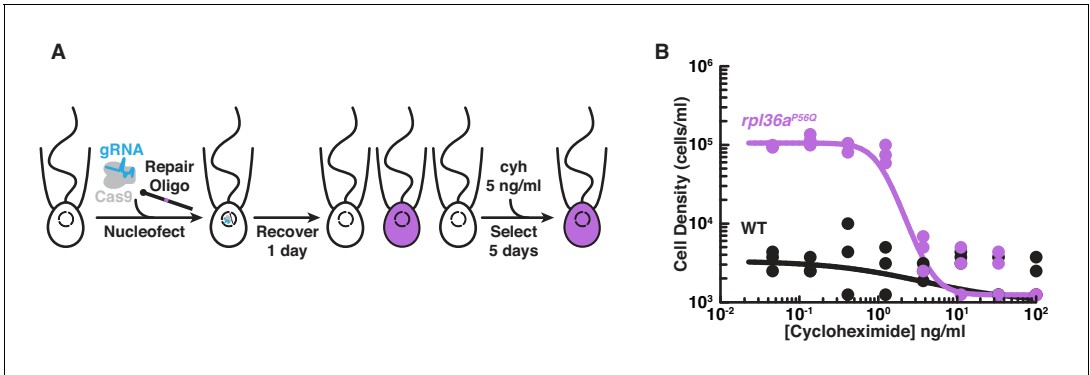

**Figure 2.** Engineered cycloheximide resistance in *S. rosetta* provides a proof-of-principle for Cas9-mediated genome editing. (A) Schematic of Cas9-mediated genome editing to engineer cycloheximide resistance in *S. rosetta*. Nucleofection was used to deliver *Sp*Cas9 (gray) bound to gRNA (cyan), which together form the *Sp*Cas9 RNP, and repair oligonucleotides (Repair Oligo; *Figure 2—figure supplement 2*) to engineer cycloheximide resistance. After recovering cells for one day, successfully edited cells were selected by growth in media supplemented with cycloheximide (cyh), which inhibits the growth of wild-type cells (*Figure 2—figure supplement 1*) and selects for cycloheximide-resistant cells (purple). (B) A designer cycloheximide-resistant allele (*Figure 2—figure supplement 2*) allows cell proliferation in the presence of cycloheximide. Wild-type (WT, black dots and line) and *rpl36a*$^{P56Q}$ (purple dots and line) strains were placed into media supplemented with a range of cycloheximide concentrations (x-axis) at a cell density of $10^4$ cells/ml and then were grown for two days. *rpl36a*$^{P56Q}$ grew to higher cell densities than the wild-type strain at cycloheximide concentrations < 10 ng/ml. At higher concentrations, cycloheximide inhibited growth of both strains. The dots show cell densities from three independent replicates. The lines show the average from independently fitting a dose inhibition curve to the cell densities from three independent experiments.

The online version of this article includes the following figure supplement(s) for figure 2:

**Figure supplement 1.** An approach for selecting cycloheximide resistance in *S. rosetta*.
**Figure supplement 2.** Engineered cycloheximide resistance establishes genome editing conditions.

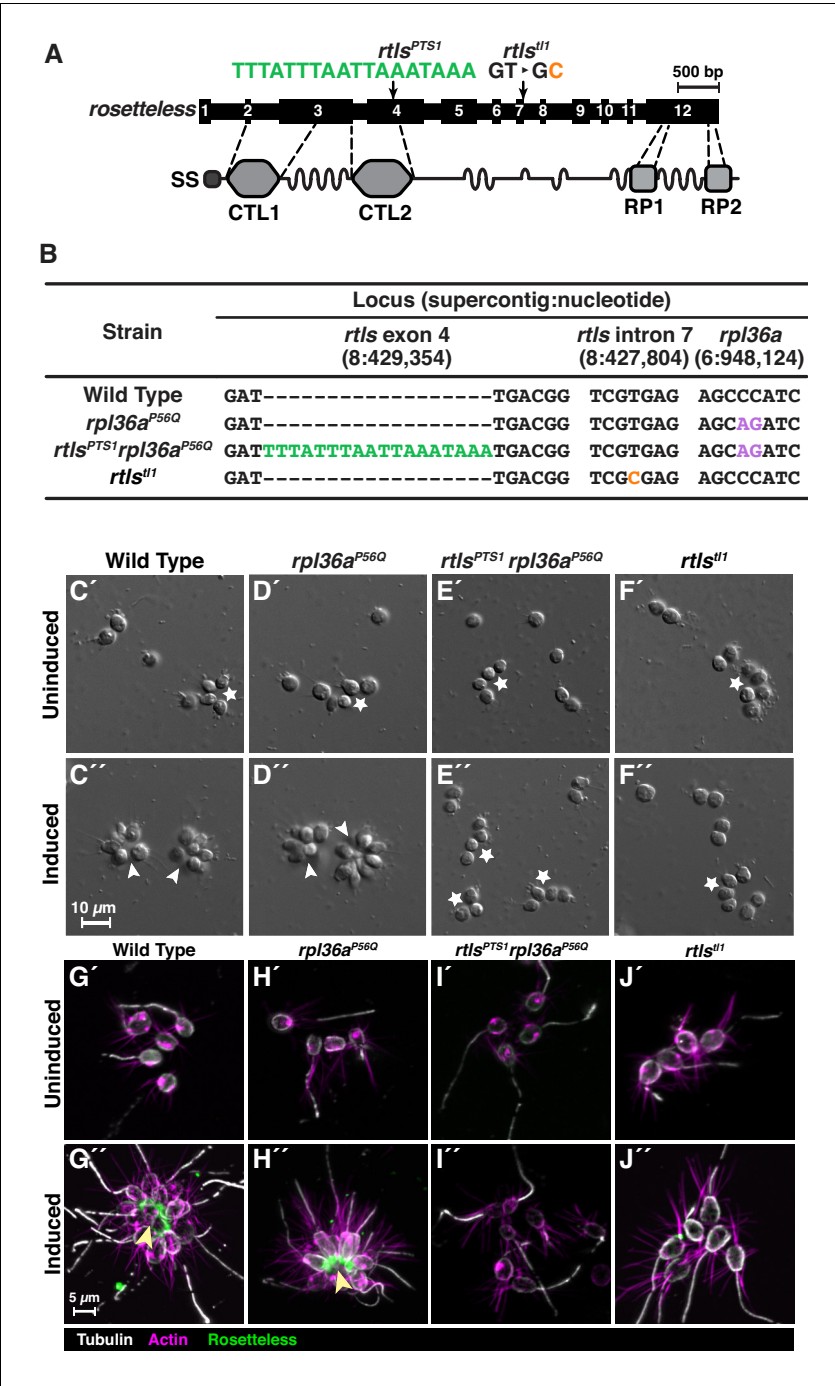

**Figure 3.** Genome editing of *rosetteless* enables targeted disruption of multicellular development in *S. rosetta*. (**A**) An engineered mutation in *rosetteless* introduces a premature termination sequence (PTS) to knockout the expression of *rosetteless*. The *rosetteless* gene (exons shown as numbered black boxes, connected by introns) encodes a secreted protein (SS denotes the signal sequence for secretion) with two C-type lectin domains (CTL1 and CTL2) and two carboxy-terminal repeats (RP1 and RP2). A forward genetic screen (*Levin et al., 2014*) identified a mutation, *rtls^{tl1}*, in which a T to C transition in the seventh intron disrupts splicing and knocks out *rosetteless* expression. To increase the likelihood of disrupting *rosetteless* function with genome editing, we designed the *rtls^{PTS1}* mutation that introduces a PTS (green), with a poly-adenylation sequence and stop codons in each reading frame, into the fourth exon of the gene. (**B**) The genotypes of strains established from genome-editing confirm that *rosetteless* and *rpl36a* incorporated the designed mutations. To enrich for genome-edited cells, *Sp*Cas9 RNPs and repair templates for introducing *rpl36a^{P56Q}* (*Figure 2—figure supplement 2*) and *rtls^{PTS1}*

*Figure 3 continued on next page*

*Figure 3 continued*

were simultaneously delivered into *S. rosetta*. Afterward, cycloheximide resistant cells were clonally isolated and screened for cells that did not develop into rosettes in the presence of RIFs. The genotypes of $rtls^{PTS1}$ $rpl36a^{P56Q}$, and $rpl36^{P56Q}$ confirmed that strains established from genome editing had the $rpl36^{P56Q}$ allele and the strain with the *rosetteless* phenotype also had the $rtls^{PTS1}$ allele. In addition, the wild-type and genome edited strains lacked the T to C transition in the 5'-splice site of intron seven that defined the $rtls^{tl1}$ allele. (C–F) Phenotypes of genome-edited strains correspond to their respective genotypes. In the absence of RIFs, all strains (C', D', E', and F') grew as chains (stars) or single cells. Upon the addition of RIFs, the wild-type (C") and $rpl36^{P56Q}$ strains (D") formed rosettes (arrowheads). In contrast, $rtls^{PTS1}$ $rpl36^{P56Q}$ (E") and $rtls^{tl1}$ (F") did not form rosettes. (G–J) Mutations in *rosetteless* prevent the secretion of Rosetteless protein at the basal end of cells and into the interior or rosettes. Immunofluorescent staining for Rosetteless (green), alpha tubulin (gray), and actin (magenta) in wild-type (G), $rpl36a^{P56Q}$ (H), $rtls^{PTS1}$ $rpl36a^{P56Q}$ (I), and $rtls^{tl1}$ (J) strains with (G"–J") and without (G'–J') rosette induction. Rosetteless localizes in the interior of rosettes (arrow) in wild-type and $rpl36a^{P56Q}$ but not $rtls^{PTS1}$ $rpl36a^{P56Q}$ and $rtls^{tl1}$.

The online version of this article includes the following figure supplement(s) for figure 3:

**Figure supplement 1.** Phenotypes of *rosetteless* mutants correspond to their genotypes.
**Figure supplement 2.** Wild-type and mutant strains proliferate similarly.

---

the disruption of *rtls* expression and rosette development (*Figures 3A* and *1C–E*; *Levin et al., 2014*). We therefore sought to generate a new *rtls* knockout allele, whose phenotype we predicted would be the loss of rosette development.

To increase the likelihood of generating a *rtls* knockout through genome editing, we aimed to introduce sequences that would prematurely terminate transcription and translation near the 5' end of the gene. First, we designed a gRNA that would target *Sp*Cas9 to the 5' end of *rtls*. Next, we designed a general-purpose premature termination sequence (PTS), an 18-base, palindromic sequence (5'-TTTATTTAATTAAATAAA-3') that encodes polyadenylation sequences and stop codons on both strands and in each possible reading frame. This sequence should prematurely terminate transcription and translation to either create a gene truncation or fully knockout target gene expression. We then designed a DNA oligonucleotide repair template in which the PTS was inserted into 100 bp of *rtls* sequence centered around the *Sp*Cas9 cleavage site (supercontig 8: 429,354 nt).

The low efficiency of transfection (~1%; *Booth et al., 2018*), the inability to select for cells with the Rosetteless phenotype, and the unknown but potentially low efficiency of genome editing meant that it might be difficult to recover cells in which *rosetteless* had been edited. To overcome this challenge, we sought to simultaneously edit *rosetteless* and *rpl36a* by transfecting cells with RNPs complexed with gRNAs and DNA repair templates for both knocking out *rosetteless* and engineering cycloheximide resistance. In other organisms, this approach has allowed for co-selection by using a selectable marker to improve the recovery of cells that contain a second mutation in a different locus. In *S. rosetta*, we found that 10.4–16.5% of cycloheximide resistant cells contained the $rtls^{PTS1}$ allele when *rosetteless* and *rpl36a* were co-edited (*Figure 3—figure supplement 1A*).

By first selecting for cycloheximide resistance and then performing clonal isolation by limiting dilution, we were able to isolate multiple clonal lines that were resistant to cycloheximide. We focused on one strain that correctly formed rosettes in response to bacterial rosette inducing factors (RIFs; *Figure 3D*; *Alegado et al., 2012*; *Woznica et al., 2016*) and two cycloheximide-resistant strains that failed to form rosettes in the presence of RIFs (representative strain shown in *Figure 3E*). Genotyping of these strains at exon 4 of *rosetteless* and at *rpl36a* (*Figure 3B*) showed that: (1) all three cycloheximide resistant strains established from the same genome-edited population had the cycloheximide resistance allele, (2) the strains that developed into rosettes only had the cycloheximide resistant allele, $rpl36a^{P56Q}$, and (3) the two strains that did not develop into rosettes also had the PTS in *rosetteless* exon 4, meaning their genotype was $rtls^{PTS1}$ $rpl36a^{P56Q}$ (*Figure 3B*). For comparison, we also genotyped wild-type, $rpl36a^{P56Q}$, $rtls^{PTS1}$ $rpl36a^{P56Q}$, and $rtls^{tl1}$ strains at intron 7 of *rosetteless*, where the $rtls^{tl1}$ mutation was mapped, to underscore that $rtls^{PTS1}$ is an independent mutation that prevents the development of rosettes (*Figure 3B*).

To further validate the genotype-to-phenotype relationship of the *rosetteless* knockouts (*Figure 3C–F*), we analyzed the percentage of cells that developed into rosettes (*Figure 3—figure supplement 1B*), the localization of the Rosetteless protein (*Figure 3G-J*), and the rates of

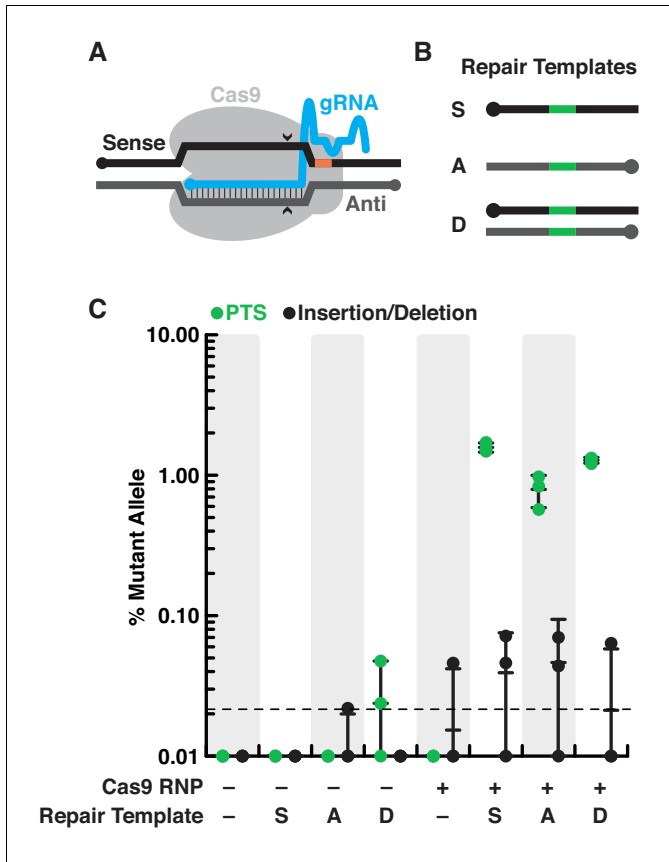

**Figure 4.** *S. rosetta* preferentially introduces genome-edited mutations from DNA templates. (**A**) Schematic of a gRNA targeting *Sp*Cas9 to a genomic locus of interest. A gRNA (cyan, knobs indicate 5' ends) that encodes a 20 nt targeting sequence from the sense strand of a genomic locus (black) hybridizes with the antisense strand (dark gray). *Sp*Cas9 (light gray) introduces a double-stranded break at the genomic locus (carets), 3 bp upstream of a protospacer adjacent motif (PAM, orange). (**B**) We designed a panel of repair oligonucleotides to test the preferred substrates for repairing double-stranded breaks introduced by *Sp*Cas9 at *rosetteles* exon 4. Oligonucleotide repair templates containing the PTS sequence (green) were delivered as single-stranded DNA in the sense (**S**) or anti-sense (**A**) orientations and as a double-stranded template (**D**) to test which most efficiently templated DNA repair at the *Sp*Cas9 cleavage site. (**C**) *Sp*Cas9 stimulated the introduction of PTS mutations from DNA templates. Repair templates with a PTS (from panel B) were delivered in the presence and absence of *Sp*Cas9 (+/–). A ~ 450 bp fragment surrounding the $rtls^{PTS1}$ cleavage site was amplified from cells that had been transfected the previous day to prepare deep sequencing libraries for quantifying the frequency of PTS insertions (green) or insertions/deletions from error prone editing (black). Each experiment was performed three independent times (dots; mean and standard deviations are shown with lines). The dotted line indicates the limit of detection of the sequencing, based on a 6-base, randomized barcode. Upon transfection with the *Sp*Cas9 RNP, 10x more mutations from repair templates (1–2%, green dots) were detected than untemplated insertions or deletions (black dots). In the absence of *Sp*Cas9, mutations generated from a double-stranded template, but not single-stranded templates, were rarely (<0.1%) and unreliably (2 of 3 trials) found.

The online version of this article includes the following figure supplement(s) for figure 4:

**Figure supplement 1.** Characterization of editing outcomes at the *rosetteless* locus with different types of repair templates.

proliferation (*Figure 3—figure supplement 2*) in the wild-type, $rpl36a^{P56Q}$, $rtls^{PTS1}$ $rpl36a^{P56Q}$, and $rtls^{tl1}$ strains of *S. rosetta*. In each of these assays, the $rtls^{PTS1}$ $rpl36a^{P56Q}$ strains exhibited the same phenotype as $rtls^{tl1}$ (*Figure 3*, compare E to F): no cells developed into rosettes (*Figure 3—figure supplement 1B*), an anti-Rosetteless antibody did not detect Rosetteless protein at the basal end of cells (*Figure 3*, compare I-J to G), and the mutant and wild-type strains proliferated comparably well (*Figure 3—figure supplement 2*). Furthermore, $rpl36a^{P56Q}$ developed into wild-type rosettes

(*Figure 3*, compare D to C and *Figure 3—figure supplement 1B*) localized Rosetteless protein to the basal end of cells (*Figure 3*, compare H to G), and proliferated as rapidly as the wild-type strain (*Figure 3—figure supplement 2A–B,E*), demonstrating that the act of genome editing alone does not yield non-specific defects in rosette development. Our ability to engineer a new *rosetteless* allele, *rtls*^PTS1^, that mimics the rosette-defect phenotype of *rtls*^tl1^ demonstrates the potential of genome editing as a general tool for generating targeted gene knockouts in choanoflagellates.

## *S. rosetta* preferentially introduces genome-edited mutations from DNA templates

Thus far, we had only detected mutations from repair templates with homology arms spanning both sides of the double-strand break (*Figures 2* and *3*, and *Figure 2—figure supplement 2*). However, selecting for cycloheximide resistance may have favored those repair outcomes, as insertion or deletion (INDEL) mutations arising from untemplated repair are likely to be deleterious for the function of *rpl36a*. Therefore, to investigate the frequency of template-mediated repair in the absence of selection, we sought to edit *rosetteless*, which is not required for viability (*Figure 3—figure supplement 2*).

As prior work has shown that editing outcomes in different cell types (*Harrison et al., 2014*; *Yeh et al., 2019*) can depend on the length and orientation (anti-sense or sense) of homology arms (*Lin et al., 2014*; *Kaulich et al., 2015*; *Richardson et al., 2016*; *Paix et al., 2017*; *Li et al., 2019*; *Okamoto et al., 2019*) and chemical modifications of DNA repair templates (*Tsai et al., 2015*; *Renaud et al., 2016*; *Yu et al., 2020*), we designed a panel of diverse double- and single-stranded DNA repair templates that all contained the PTS (*Figure 4B* and *Figure 4—figure supplement 1*). The double-stranded templates contained phosphorylation or phosphorothioate bonds at their 5' ends (*Figure 4—figure supplement 1A*); whereas, the single-stranded templates varied in their orientation and presence of 5' or 3' homology arms (*Figure 4—figure supplement 1B*). We transfected cells with these repair templates with or without the *Sp*Cas9 RNP. After the cells recovered for one day, we amplified a ~ 450 bp fragment around the *Sp*Cas9 cut site for deep sequencing (*Yang et al., 2013*; *Lin et al., 2014*) and quantified the frequency and type of mutation after genome editing.

We found that the *S. rosetta* genome could be edited in a *Sp*Cas9-dependent manner using a variety of templates (*Figure 4C*, *Figure 4—figure supplement 1C*). In the presence of the *Sp*Cas9, INDEL mutations occurred at a frequency of <0.1%, either in the presence or absence of DNA repair templates (*Figure 4C*, *Figure 4—figure supplement 1D–E*). In contrast, the addition of *Sp*Cas9 with DNA repair templates that encoded the PTS resulted in PTS mutations at a frequency of 0.79–1.57%, which is at a ten-fold higher frequency than the INDEL mutational frequency (Two-Factor ANOVA: $p<10^{-13}$). Notably, the total frequency of genome editing (~1%) is on the same order of magnitude as transfection efficiency (~1%; *Booth et al., 2018*), suggesting that the delivery of *Sp*Cas9 and repair templates is the biggest factor limiting genome editing efficiency.

In the absence of *Sp*Cas9, we observed two types of *Sp*Cas9-independent genome edits. The first was a single INDEL mutation detected in a population of cells transfected with the antisense repair template, and the second was the detection of PTS mutations at an average frequency of ~0.02% upon the delivery of a double-stranded repair template. Although the frequency of these mutations occurred at a rate less than or equal to the detection threshold (~0.02%), meaning that we could not confidently conclude that differences exist between any of the samples (Two-Factor, ANOVA: $p<0.27$), the presence of these mutations is consistent with a low frequency of endogenous DNA repair. These results also emphasize that the addition of *Sp*Cas9 was essential for efficient, targeted mutagenesis (Two-Factor ANOVA: $p<0.02$ for INDEL mutations and $p<10^{-13}$ for PTS mutations).

Altogether, our optimization efforts revealed that the delivery of the *Sp*Cas9 with a DNA template spanning both sides of the *Sp*Cas9 cleavage site introduced PTS mutations at a frequency of ~1%. We recommend using a sense-oriented, single-stranded template for genome editing, as this template led to the highest frequency of PTS mutations (Single-Factor ANOVA, Tukey Multiple Comparison Test: *p<0.04*) and costs less to synthesize than a double-stranded template.

## Discussion

The establishment of Cas9-mediated genome editing advances *S. rosetta* as a model for illuminating the evolution of development in choanoflagellates and their closest living relatives, animals. We were able to overcome initial failed efforts to establish genome editing in *S. rosetta* by engineering cycloheximide resistance in *rpl36a* as a selectable marker, similar to the use of selectable markers during the establishment of genome editing in other eukaryotes, including Fungi (*Foster et al., 2018*), green algae (*Ferenczi et al., 2017*), and nematodes (*Arribere et al., 2014*; *Kim et al., 2014a*; *Ward, 2015*). Single-copy ribosomal protein genes like *rpl36a* offer certain advantages for engineering drug resistance markers with genome editing. First, resistance mutations in ribosomal protein genes have been genetically and biochemically characterized for a variety of drugs in diverse eukaryotes (*Sutton et al., 1978*; *Ares and Bruns, 1978*; *Kawai et al., 1992*; *Dehoux et al., 1993*; *Kondo et al., 1995*; *Kim et al., 1998*; *Stevens et al., 2001*; *Garreau de Loubresse et al., 2014* and references therein). In our case, interpreting alignments among Rpl36a sequences from *S. rosetta* and organisms in the context of structures of eukaryotic ribosomes provided a starting point for customizing cycloheximide resistant alleles, a strategy that can also extend to other organisms. Second, the specificity of antibiotics that inhibit eukaryotic or prokaryotic translation can be leveraged to tailor genetic tools for particular organisms in complex communities. For example, cycloheximide binds selectively to eukaryotic ribosomes, resulting in the inhibition of *S. rosetta* growth and not that of its food source: live prey bacteria. Combining these advantages to establish genome editing in *S. rosetta* provided the first proof-of-principle for genome editing and allowed us to characterize the essential parameters before targeting other genes.

With the newfound potential for reverse genetics, we revisited the genetic basis of multicellular rosette development in *S. rosetta*. A previous forward genetic screen followed by mapping crosses implicated the C-type lectin gene *rosetteless* in the regulation of rosette development (*Levin et al., 2014*). At the time, however, it was not possible to independently corroborate *rosetteless* function with targeted mutations. In this study, we used genome editing to introduce a premature termination sequence in *rosetteless* and found that strains with the engineered *rosetteless* mutation have the same rosette defect phenotype as cells with the original *rtls^{tl1}* mutation, demonstrating that *rosetteless* is necessary for rosette development.

Moving forward, the approach established here promises to accelerate future research on choanoflagellates. It will now be possible for choanoflagellate researchers to introduce candidate mutations into a wild-type strain or correct the causative mutations in the original mutant strain to cleanly test the connection between genotype and phenotype. Similarly, for reverse genetics, the use of different guide RNAs and repair templates through CRISPR/*Sp*Cas9 genome editing will allow researchers to engineer multiple independent alleles to better understand the phenotype of targeted gene knockouts. Tools that have been previously used in forward genetic approaches, such as genetic crosses (*Levin and King, 2013*; *Levin et al., 2014*; *Woznica et al., 2017*) and stable transgenesis (*Wetzel et al., 2018*), may also provide the means to rapidly generate strains with different genetic backgrounds to complement mutants, to reveal epistasis, or simply to eliminate off-target mutations that may arise during genome editing.

Importantly, the establishment of genome editing in *S. rosetta* offers the first model choanoflagellate to investigate the ancestral and core functions of genes that evolved as integral regulators of animal biology. The *S. rosetta* genome (*Fairclough et al., 2013*) encodes receptors for immunity (e.g. Toll-like receptors), intercellular communication (e.g. receptor tyrosine kinases), and adhesion (e.g. cadherins, C-type lectins, and immunoglobulins) as well as master regulators of cell differentiation (e.g. forkhead, homeodomain, p53 and sox transcription factors). As a simple microbial model, *S. rosetta* now may serve as an accessible system for uncovering the conserved functions of genes that are not as readily studied in the more complex context of multicellular animals. Moreover, *S. rosetta* is just one tip on the choanoflagellate branch. Recent surveys of 21 choanoflagellate transcriptomes and genomes from uncultured species have revealed that choanoflagellates are at least as genetically diverse as animals (*Richter et al., 2018*; *López-Escardó et al., 2019*), with other species retaining genetic pathways or exhibiting behaviors that are not found in *S. rosetta* (e.g., *Marron et al., 2013*; *Leadbeater, 2015*; *Brunet et al., 2019*). Together with future findings from *S. rosetta*, we anticipate that the establishment of genome editing in other choanoflagellates will

provide an increasingly complete portrait of the last common ancestor of choanoflagellates and animals.

## Materials and methods

### Culturing choanoflagellates

Strains of *S. rosetta* were co-cultured with *Echinicola pacifica* bacteria (*Levin and King, 2013*); American Type Culture Collection [ATCC], Manassas, VA; Cat. No. PRA-390) in seawater-based media enriched with glycerol, yeast extract, and peptone to promote the growth of *E. pacifica* that serve as the choanoflagellate prey (*Levin and King, 2013*; *Booth et al., 2018*). We further supplemented this media with cereal grass (*King et al., 2009*; *Fairclough et al., 2010*; Carolina Biological Supply Company, Burlington, NC; Cat. No. 132375), which we call high nutrient media (*Supplementary file 1*-Table A), as we noticed that this addition promoted *S. rosetta* growth to a higher cell density (~$10^7$ cells/ml [*Figure 3—figure supplement 2A*] versus ~$10^6$ cells/ml (*Booth et al., 2018*). To maintain rapidly proliferating cells in an abundance of nutrients, cultures were diluted 1 in 30 daily or 1 in 60 every two days into 6 ml of high nutrient media in 25 $cm^2$ vented culture flasks (Corning, Oneonta, NY, USA; Cat. No. 430639) and incubated at 22°C and 60% relative humidity. To prevent an overgrowth of bacteria when *S. rosetta* experienced stress, such as after transfections or during clonal isolation, we cultured *S. rosetta* in low nutrient media, which is 0.375x high nutrient media (*Supplementary file 1*-Table A).

### Purification of outer membrane vesicles that contain RIFs

Rosette inducing factors (RIFs) contained in outer membrane vesicles (OMVs) from *Algoriphagus machipongonensis* (*Alegado et al., 2013*; ATCC; Cat. No. BAA-2233) can be used to induce rosette development in *S. rosetta* (*Alegado et al., 2012*; *Woznica et al., 2016*). *A machipongonensis* OMVs were purified using the protocol in *Woznica et al., 2016*. In summary, a 200 ml culture of 25x high nutrient media without cereal grass was inoculated from a single colony of *A. machipongonensis* and grown in a 1 l, baffled flask by shaking at 200 rpm for 3 days at 30°C. Afterwards, the bacteria were pelleted in 50 ml conical tubes by centrifugation at 4500 g and 4°C for 30 min. The pellet was discarded and the supernatant was filtered through a 0.22 µm vacuum filter. Outer membrane vesicles were pelleted from the filtered supernatant by ultracentrifugation at 36,000 g and 4°C in a fixed-angle, Type 45 Ti rotor (Beckman Coulter Life Sciences, Indianapolis, IN; Cat. No. 339160) for 3 hr. After discarding the supernatant, the pellet of outer membrane vesicles, which has an orange hue, was resuspended in a minimal volume of 50 mM HEPES-KOH, pH 7.5 and then incubated at 4°C overnight to fully dissolve the pellet. Last, the pellet was sterile filtered through a 0.45 µm polyvinylidene fluoride syringe filter (Thermo Fisher Scientific, Waltham, MA; Cat. No. 09-720-4) into a sterile tube.

 The rosette-inducing activity of the OMVs was tested by serially diluting the purified OMVs in a 24-well plate, with each well containing 0.5 ml of *S. rosetta* at a concentration of $10^4$ cells/ml and *E. pacifica*. The cells were incubated with OMVs at 22°C for 48 hr and then fixed with formaldehyde before counting the fraction of cells (*n* = 100) in rosettes. The dilution of lipids in which half of *S. rosetta* cells formed rosettes was defined as two unit/ml. All subsequent rosette inductions were performed with OMVs at a final concentration of 10 units/ml.

### Genome editing

Below we describe the considerations for the design and preparation of gRNAs and repair oligonucleotides for genome editing. The particular gRNAs and DNA repair template sequences for each given experiment are provided in *Supplementary file 1*-Table B.

### Design and preparation of gRNAs

Upon inspecting the structure of the *Sp*Cas9 RNP poised to cleave a DNA target (*Jiang et al., 2016*), we concluded that sequences adjacent to and upstream of the PAM sequence (5′-NGG-3′), which have been reported to bias *Sp*Cas9 activity in vivo (*Doench et al., 2014*; *Wu et al., 2014*; *Xu et al., 2015*; *Moreno-Mateos et al., 2015*; *Horlbeck et al., 2016*; *Liu et al., 2016*; *Kaur et al., 2016*; *Gandhi et al., 2017*), likely influence *Sp*Cas9 recognition by stabilizing the conformation of

the DNA target for cleavage. Therefore, we accounted for biases in *Sp*Cas9 recognition by choosing gRNAs sequences that conformed, as much as possible, to the motif 5'-HNNGR<u>SGG</u>H-3', in which the PAM is underlined, N stands for any base, R stands for purine, S stands for G or C, and H stands for any base except G. This motif was first used to search for suitable targets (*Peng and Tarleton, 2015*) in cDNA sequences. We reasoned that initially searching for putative targets in cDNA sequences would ensure that gRNAs direct *Sp*Cas9 to cleave in protein coding regions of genes, and we later verified that putative gRNAs recognized genomic sequences instead of exon-exon junctions. Finally, we filtered out putative gRNA sequences with potential secondary sequences that can impede gRNA hybridization with DNA targets (*Thyme et al., 2016*) by evaluating their predicted secondary structures (*Lorenz et al., 2011*) and keeping gRNAs with predicted folding free energies greater than −1.5 kcal/mol.

gRNAs were prepared by annealing synthetic CRISPR RNA (crRNA) with a synthetic trans-activating CRISPR RNA (tracrRNA). The synthetic crRNA contains the 20 nt sequence for gene targeting and an additional sequence to anneal to the tracrRNA that binds to *Sp*Cas9. Alternatively, we also performed genome editing (*Figure 2* and *Figure 2—figure supplement 2H*) with in vitro transcribed gRNAs (see below) that link the crRNA and tracrRNA into one continuous strand (*Jinek et al., 2012*; *Chen et al., 2013*), but we found that genome editing with crRNA/tracrRNA was the most time- and cost-effective. To prepare a functional gRNA complex from synthetic RNAs, crRNA and tracrRNA (Integrated DNA Technologies [IDT], Coralville, IA, USA) were resuspended to a final concentration of 200 μM in duplex buffer (30 mM HEPES-KOH, pH 7.5; 100 mM potassium acetate; IDT, Cat. No. 11-01-03-01). Equal volumes of crRNA and tracrRNA stocks were mixed together, incubated at 95°C for 5 min in an aluminum heat block, and then the entire heat block was placed at room temperature to slowly cool the RNA to 25°C. The RNA was stored at −20°C.

## In vitro transcription of gRNAs

DNA templates for in vitro transcription of gRNAs were amplified by PCR (Q5 DNA Polymerase; New England Biolabs [NEB], Ipswich, MA, USA, Cat. No. M0491L) from synthetic DNA templates (IDT; *Supplementary file 1*-Table B) that had a T7 promoter sequence appended to the 5' end of the guide sequence and a trans-activating CRISPR RNA (tracrRNA) sequence (*Chen et al., 2013*) at the 3' end. The purified DNA templates (PCR cleanup kit; Qiagen, Venlo, NLD; Cat. No. 28006) were used to synthesize gRNAs with T7 RNA polymerase (*Milligan and Uhlenbeck, 1989*) in reactions set up with these components: 40 mM Tris-HCl, pH 8.0; 2.5 mM spermidine; 0.01% (v/v) Triton X-100; 5 mM GTP; 5 mM UTP; 5 mM ATP; 5 mM CTP; 80 mg/ml PEG 8000; 32 mM magnesium chloride; 5 mM dithiothreitol; 10 ng/μl template DNA; 0.5 U/μl SUPERase•In (Thermo Fisher Scientific, Waltham, MA; Cat. No. AM2696); 2 U/μl T7 RNA polymerase (Thermo Fisher Scientific, Cat. No. EP0113); 0.025 mg/ml pyrophosphatase (Thermo Fisher Scientific, Cat. No. EF0221). After incubating the transcription reaction at 37°C for >4 hr, the DNA template was digested with the addition of 0.1 U/μl TURBO DNase (Thermo Fisher Scientific, Cat. No. AM2239). After assessing the transcription products on denaturing, urea-polyacrylamide gel electrophoresis (PAGE), we found that the in vitro transcriptions yielded high amounts of gRNA with few byproducts. Therefore, we used a simplified protocol to purify gRNAs by first removing contaminating nucleotides with a desalting column (GE Healthcare Lifesciences, Pittsburgh, PA; Cat. No. 17085302) to exchange gRNA into 1 mM sodium citrate, pH 6.4. The gRNAs were then precipitated from the solution by adding 0.25 volumes of RNA precipitation buffer (1.2 M sodium acetate, pH 5.2; 4 mM EDTA-NaOH, pH 8.0; 0.04% sodium dodecyl sulfate [SDS]) and 2.5 volumes of ethanol. The precipitated RNA was centrifuged for 60 min at 4°C, washed once with 70% ethanol/water, and finally resuspended in 1 mM sodium citrate, pH 6.4.

After determining the concentration of gRNA, which has a 260 nm extinction coefficient of $1.41 \times 10^6$ M$^{-1}$cm$^{-1}$, by UV-vis spectroscopy, the gRNA was diluted to a final concentration of 50 μM with 1 mM sodium citrate, pH 6.4. To ensure that the gRNA was properly folded, the gRNA was placed at 95°C for 5 min in an aluminum heat block and then slowly cooled to 25°C by placing the aluminum block on a room temperature bench top. Finally, gRNAs were stored at −20°C.

## Design and preparation of repair oligonucleotides

Repair oligonucleotides for generating knockouts were designed by copying the sequence 50 bases upstream and downstream of the *Sp*Cas9 cleavage site, which itself is 3 bp upstream of the PAM sequence (for example, 5'-N-cleave-NNN<u>NGG</u>-3'; PAM sequence underlined). A PTS (5'-TTTA TTTAATTAAATAAA-3') was inserted at the cleavage site. Importantly, this sequence has a stop codon (TAA) in each possible reading frame to terminate translate, a polyadenylation sequence (AATAAA) to terminate transcription, and a PacI sequence (5'-TTAATTAA-3') that can be used to genotype with restriction digests. Moreover, the knockout sequence is palindromic, so it can be inserted in the sense or antisense direction of a gene and still generate a knockout.

Dried oligonucleotides (IDT) were resuspended to a concentration of 250 μM in a buffer of 10 mM HEPES-KOH, pH 7.5, incubated at 55°C for 1 hr, and mixed well by pipetting up and down. The oligonucleotides were stored at −20°C.

## SpCas9 expression and purification

For efficient genome editing, we purified or purchased (NEB, Cat. No. M0646M) an engineered version of *Streptomyces pyogenes* Cas9 that has SV40 nuclear localization sequences (NLS) at the amino- and carobxy- termini of *Sp*Cas9. Below we describe a simplified purification procedure based on the previously published work (*Jinek et al., 2012*).

### Vector construction

Using a variation of Gibson cloning (*Gibson et al., 2009*; NEB, Cat. No. E2621L), we modified a vector (*Jinek et al., 2012*); Addgene, Watertown, MA; Cat. No. 69090) for expressing *Sp*Cas9 in *Escherichia coli* by inserting tandem SV40 NLSs at the amino terminus of *Sp*Cas9. A similar construct (Addgene, Cat. No. 88916) has been shown to increase nuclear localization in mammalian cells (*Cong et al., 2013*; *Staahl et al., 2017*). The expression vector has a hexahistidine (His$_6$) tag and maltose binding protein (MBP) fused to the amino terminus of *Sp*Cas9. A tobacco etch virus (TEV) protease cleavage site between MBP and the amino terminal nuclear localization sequence on Cas9 facilitates the removal of the His$_6$-MBP tag from *Sp*Cas9.

### Protein expression

The *Sp*Cas9 expression vector was transformed into the BL21 Star (DE3) strain of *E. coli* (Thermo Fisher Scientific, Cat. No. C601003), and a single colony of the transformants was inoculated into Miller's LB broth (*Atlas, 2010*) for growing a starter culture overnight at 37°C with shaking at 200 rpm. 20 ml of the starter culture was diluted into 1 L of M9 medium (*Atlas, 2010*) and the culture was grown at 37°C with shaking at 250 rpm until the OD$_{600}$ = 0.60. At that cell density, the culture was shifted to 16°C for 15 min and *Sp*Cas9 expression was induced by addition isopropyl β-D-1-thio-galactopyranoside (IPTG) to a final concentration of 0.5 mM. The culture was grown at 16°C over-night and cells were harvested by centrifugation at 4900 g and 4° for 15 min in a swinging bucket centrifuge. The supernatant was discarded and the bacterial pellets were flash frozen in liquid nitro-gen and stored at −80°C.

### Protein purification

The bacterial pellet was lysed by resuspending 1 g of bacterial pellet in 9 ml of lysis buffer (150 mM potassium phosphate, pH 7.5; 500 mM sodium chloride; 5 mM imidazole, pH 8.0; 1 mM Pefabloc SC; 2 mM 2-mercaptoethanol; 10% [v/v] glycerol; one protease inhibitor tablet [cOmplete, EDTA-free; Roche; Cat. No, 04693132001] per 20 ml of lysate) and lysing with a microfluidizer. The lysate was centrifuged at 30,000 g and 4°C for 30 min to remove insoluble debris.

The supernatant was passed through an Ni-NTA Agarose column (Qiagen, Cat. No. 30210) equili-brated in elution buffer (150 mM potassium phosphate, pH 7.5; 500 mM sodium chloride; 5 mM imidazole, pH 8.0; 2 mM 2-mercaptoethanol; 10% [v/v] glycerol), using 1 ml of resin per 10 grams of bacterial pellet. The column was washed with 10 column volumes (CV) of lysis buffer, 5 CV of elution buffer supplemented with 10 mM imidazole, and 3 CV of wash elution buffer supplemented with 20 mM imidazole. The protein was eluted from the column with 4 CV of elution buffer supplemented with 240 mM imidazole. After determining the protein concentration by UV-vis spectroscopy (using an extinction coefficient of 0.18829 μM$^{-1}$cm$^{-1}$ for *Sp*Cas9), TEV protease was added at 1:20 molar

ratio of TEV protease to *Sp*Cas9. SpCas9 supplemented with TEV protease was placed in a dialysis bag with a 3500 dalton molecular weight cut-off (MWCO) and dialyzed against dialysis buffer (100 mM potassium phosphate, pH 7.5; 2 mM 2-mercaptoethanol; 10 mM imidazole; 10% [v/v] glycerol) overnight at 4°C. Afterwards, the dialyzed protein was passed over the Ni-NTA column that had been equilibrated in dialysis buffer to remove His$_6$-MBP tag from *Sp*Cas9, which is in the flow through. The flow through was loaded onto HiTrap SP High-Performance (GE Healthcare Lifesciences, Cat. No. 17-1152-01) column that had been equilibrated in dialysis buffer. The column was extensively washed with dialysis buffer prior to eluting the protein in S-elution buffer (500 mM potassium phosphate, pH 7.5; 3 mM dithiothreitol; 0.3 mM EDTA-KOH, pH 8.0; 10% [v/v] glycerol). The purity was evaluated by SDS-PAGE, and the concentration was measured using UV-vis spectroscopy. Afterwards, the purified protein was concentrated with a 100,000 MWCO centrifugal filter to a final concentration of 20–25 µM. The concentrated protein was flash-frozen in liquid nitrogen and stored at −80°C.

## Delivery of gene editing cargoes with nucleofection

SpCas9 RNPs and DNA repair templates were delivered into *S. rosetta* using a modified method for nucleofection (*Booth et al., 2018*). Here we describe the complete transfection procedure and provide a publicly-accessible protocol specific for genome editing (https://dx.doi.org/10.17504/protocols.io.89fhz3n):

### Cell culture

Two days prior to transfection, 120 ml of high nutrient media was inoculated with a culture of *S. rosetta/E. pacifica* to a final concentration of *S. rosetta* at 8000 cells/ml. The culture was grown in a 3-layer flask (Corning; Cat. No. 353143), which has a surface area of 525 cm$^2$, at 22°C and 60% humidity.

### Assembly of Cas9/gRNA RNP

Before starting transfections, the *Sp*Cas9 RNP was assembled. For one reaction, 2 µl of 20 µM *Sp*Cas9 (NEB, Cat. No. M0646M or purified as described above) was placed in the bottom of a 0.25 ml PCR tube, and then 2 µl of 100 µM gRNA was slowly pipetted up and down with *Sp*Cas9 to gently mix the solutions. The mixed solution was incubated at room temperature for 1 hr, which is roughly the time to complete the preparation of *S. rosetta* for priming (see below).

### Thaw DNA oligonucleotides

Before using oligonucleotides in nucleofections, the oligonucleotides (prepared as above) were incubated at 55°C for 1 hr during the assembly of the *Sp*Cas9 RNP to ensure that they were fully dissolved.

### Cell washing

*S. rosetta* cells were first prepared for nucleofection by washing away feeder bacteria. The 120 ml culture started two days previously was homogenized by vigorous shaking and then split into 40 ml aliquots in 50 ml conical tubes. The aliquots were vigorously shaken before centrifuging the cells for 5 min at 2000 g and 22°C in a swinging bucket rotor. All but 2 ml of the supernatant, which remains cloudy with *E. pacifica* bacteria, was gently pipetted off of the pellet with a serological pipette; a fine tip transfer pipette gently removed the remaining liquid near the pellet. The three cell pellets were resuspended in artificial seawater (ASW; see *Supplementary file 1*-Table A) for a total volume of 50 ml, combined into one conical tube, and vigorously shaken to homogenize the cells. For a second time, the resuspended cells were centrifuged for 5 min at 2000 g and 22°C. The supernatant was removed as before, the pellet was resuspended in 50 ml of artificial seawater, and the cells were homogenized by vigorous shaking. The cells were centrifuged for a third time for 5 min at 2200 g and 22°C. After removing the supernatant as described above, the cell pellet was resuspended in 400 µl of ASW. The concentration of cells was determined by diluting 2 µl of cells into 196 µl of ASW. The diluted cells were fixed with 2 µl of 37.5% formaldehyde, vortexed, and then pipetted into a fixed chamber slide for counting with Luna-FL automated cell counter (Logos Biosystems,

Anyang, KOR; Cat. No. L20001). After determining the cell concentration, the washed *S. rosetta* cells were diluted to a final concentration of $5 \times 10^7$ cell/ml and split into 100 µl aliquots.

## Priming

To prime *S. rosetta* cells for nucleofection, we treated them with a cocktail that removes the extracellular matrix as follows. Aliquots of washed cells were pelleted at 800 g and 22°C for 5 min. The supernatant was gently removed with gel-loading tips and each pellet was resuspended in 100 µl of priming buffer (40 mM HEPES-KOH, pH 7.5; 34 mM lithium citrate; 50 mM l-cysteine; 15% [wt/vol] PEG 8000; and 1 µM papain [Millipore Sigma, St. Louis, MO; Cat. No. P3125-100MG]). After incubating cells for 30–40 min, 10 µl of 50 mg/ml bovine serum albumin was added to each aliquot of primed cells to quench proteolysis from the priming buffer. Finally, the cells were centrifuged at 1250 g and 22°C for 5 min, the supernatant was removed, and the pellet was resuspended in 25 µl of SF Buffer (Lonza, Basel, Switzerland; Cat. No. V4SC-2960). The resuspended cells were stored on ice while preparing nucleofection reagents.

## Nucleofection

Each nucleofection reaction was prepared by adding 16 µl of ice-cold SF Buffer to 4 µl of the *Sp*Cas9 RNP that was assembled as described above. (For reactions that used two different gRNAs, each gRNA was assembled with *Sp*Cas9 separately and 4 µl of each RNP solution was added to SF buffer at this step). 2 µl of the repair oligonucleotide template was added to the *Sp*Cas9 RNP diluted in SF buffer. Finally, 2 µl of primed cells were added to the solution with *Sp*Cas9 RNP and the repair template. The whole solution, which has a total volume of 24 µl (30 µl for two different *Sp*Cas9 RNPs and repair templates), was placed in one well of a 96-well nucleofection plate. The well was pulsed in a Lonza shuttle nucleofector (Lonza, Cat. No. AAF-1002B and AAM-1001S) with the CM156 pulse.

## Recovery

Immediately after transfection, 100 µl of ice-cold recovery buffer (10 mM HEPES-KOH, pH 7.5; 0.9 M sorbitol; 8% [wt/vol] PEG 8000) was added to each transfection and gently mixed by firmly tapping the side of the plate or cuvette. After the cells rested in recovery buffer at room-temperature for 5 min, the whole volume of a nucleofection well was transferred to 2 ml of low nutrient media in one well of a six well plate. After 30 min, 10 µl of 10 mg/ml *E. pacifica* (prepared by resuspending a frozen 10 mg pellet of *E. pacifica* in ASW) was added to each well and the six well plate was incubated at 22°C and 60% relative humidity for downstream experiments.

## Establishing clonal strains

Here we describe how to isolate clones to establish strains. For a complete list of strains used in this study, see *Supplementary file 1*-Table C.

## Cycloheximide selection

One day after transfecting *S. rosetta* with *Sp*Cas9 RNPs repair oligonucleotides for *rpl36a*$^{P56Q}$ (*Figure 2*), 10 µl of 1 µg/ml cycloheximide was added to a 2 ml culture of transfected cells. The cells were incubated with cycloheximide for 5 days prior to genotyping and clonal isolation.

## Clonal isolation

To prepare cells for clonal isolation by limiting dilution, the initial cell density was determined by fixing a 200 µl sample of cells with 5 µl of 37.5% (w/v) formaldehyde and then by counting the fixed cells with a hemocytometer (Hausser Scientific, Horsham, PA; Cat. No. 1475) or Luna-FL automated cell counter. The cells were by diluted to a final concentration of 3 cells/ml in low nutrient sea water and then distributed in a 96 well plate with 100 µl/well. Thus, the mean frequency of finding a cell in each well is 0.3, which, according to a Poisson distribution, corresponds to a > 99% probability that a given well with *S. rosetta* was founded from a single cell. Cells were grown in a 96 well plate for 5–7 days at 22°C and 60% relative humidity. Lastly, the plate was screened using phase contrast microscopy to identify wells with *S. rosetta*. Finally, larger cultures of high nutrient media were inoculated with clonal isolates to establish strains.

## Genotyping by Sanger sequencing (*Figures 2–3* , *Figure 2—figure supplement 2*; *Figure 3—figure supplement 1*)

Cells were harvested for genotyping by centrifuging 1 ml of cells at 4250 g and 22°C for 5 min. The supernatant was removed with a fine tip transfer pipette. (Optional: To remove lingering DNA from cells that die in the course of cycloheximide selection, the pellet was resuspended in 50 µl DNase buffer [10 mM Tris-HCl, pH 7.6; 1 M sorbitol; 2.5 mM magnesium chloride; 0.5 mM calcium chloride; 0.1 U/µl Turbo DNase (Thermo Fisher Scientific; Cat. No. AM2238)] and incubated at room temperature for 30 min. Afterwards, the cells were centrifuged as before, discarding the supernatant.) The cell pellet was dissolved in 100 µl of DNAzol Direct (20 mM potassium hydroxide, 60% [w/v] PEG 200, pH 13.3–13.5; Molecular Research Center, Inc, Cincinnati, OH; Cat. No. DN131). 5 µl of the dissolved cells were added to a 50 µl PCR reaction (Q5 DNA polymerase, NEB; see *Supplementary file 1*-Table B for primer sequences) and amplified with 36 rounds of thermal cycling. Samples dissolved in DNAzol direct can be directly added to PCR reactions because the pH of DNAzol Direct dramatically drops upon a ten-fold or greater dilution (*Chomczynski and Rymaszewski, 2006*). The PCR product was purified using magnetic beads (*Oberacker et al., 2019*) and then submitted for Sanger sequencing (UC Berkeley DNA Sequencing facility).

## Cell proliferation assays (*Figure 3—figure supplement 2*)

We characterized the cell proliferation rates of *S. rosetta* strains by monitoring the concentration of cells over time to fit logistic growth curves and determine the doubling time. Cell proliferation assays started by diluting cultures to a concentration of $10^4$ cells/ml in high nutrient media and then distributing 0.5 ml of culture into each well of a 24 well plate. Every ~12 hr, the entire contents of one well were thoroughly homogenized by pipetting up and down and then transferred to a 1.5 ml conical tube. Three independent wells were taken for triplicate measures of cell concentration at every time point. The cells were fixed with 20 µl of 37.5% formaldehyde and mixed by vortexing. The fixed cells were stored at 4°C until the sample was used for determining the cell concentration after the full growth course.

The cell concentration was determined by counting the number of cells in a fixed-volume imaging chamber. In detail, the fixed cells were thoroughly homogenized by vortexing for 10 s and then pipetted up and down before transfer into a chamber of a Smart Slide (ibidi USA, Inc, Firchburg, WI; Cat. No. 80816) that has a fixed height of 200 µm. After allowing cells to settle to the bottom for 5 min, each chamber was imaged on an Axio Observer.Z1/7 Widefield microscope (Carl Zeiss AG, Oberkochen, Germany) and recorded with a Hamamatsu Orca-Flash 4.0 LT CMOS Digital Camera (Hamamatsu Photonics, Hamamatsu City, Japan) using either phase contrast for 10x (objective), in-focus images or a 20x brightfield image with a 1 µm overfocus to make the cells appear dark on a light gray background. The volume for each image was calculated from the image area, which was calibrated on the microscope, and the fixed height of the imaging chamber: $3.54 \times 10^{-4}$ ml when imaged at 10x and $8.86 \times 10^{-5}$ ml when imaged at 20x. Using automated particle detection in Fiji (*Schindelin et al., 2012*), cells were counted in each 20x image by thresholding the image to make cells appear as black spots on a white background and then each circular spot was counted with the 'Analyze Particles' function. For early time points with fewer numbers of cells, we manually counted cells in 10x images to include more cells in a greater area for a more accurate count.

Each time course was fit by least absolute deviation curve fitting to the logistic equation:

$$P_t = \frac{K \bullet P_0}{((K - -P_0) \bullet e^{-\frac{t}{T}}) + P_0},$$

where $P_t$ is the cell density at time (*t*), *K* is the carrying capacity, $P_0$ is the initial cell density, and *T* is the doubling time.

## Live-cell microscopy (*Figures 1* and *3*)

Glass-bottomed dishes (World Precision Instruments, Sarasota, FL; Cat. No. FD35-100) were prepared for imaging by covering the bottom with 500 µl of 0.1 mg/ml poly-D-lysine (Millipore Sigma; Cat. No. P6407-5MG) and incubating for 15 min. The poly-D-lysine was removed and then the dish was washed three times with 500 µl of ASW. Cells were placed into the dish by gently pipetting 500 µl of cells with a wide pipette tip.

Differential interference contrast (DIC) microscopy images were captured with a Zeiss Axio Observer.Z1/7 Widefield microscope with a Hamamatsu Orca-Flash 4.0 LT CMOS Digital Camera (Hamamatsu Photonics, Hamamatsu City, Japan) and 40×/NA 1.1 LD C-Apochromatic water immersion, 63×/NA1.40 Plan-Apochromatic oil immersion, or 100 × NA 1.40 Plan-Apochromatic oil immersion objectives (Zeiss).

## Immunofluorescent staining and imaging (*Figure 1* and *Figure 3—figure supplement 1*)

200 µl of *S. rosetta* cells were gently pipetted into chamber slides (ibidi; Cat. No.80826) coated with poly-D-lysine (see live cell imaging for coating procedure). Importantly, cells were pipetted using a tip that had been trimmed to create a larger bore for reducing shear forces. The cells were incubated on the coverslip for 30 min to allow the cells to adsorb to the surface.

Cells were fixed by adding 200 µl of 6% acetone in cytoskeleton buffer (10 mM MES, pH 6.1; 138 KCl, 3 mM MgCl$_2$; 2 mM ethylene glycol-bis($\beta$-aminoethylether)-*N*,*N*,*N'*,*N'*-tetraacetic acid [EGTA]; 600 mM sucrose) and then incubated for 10 min at room temperature. After removing 200 µl from the chamber, 200 µl of 4% formaldehyde diluted in cytoskeleton buffer was added to the chamber and then incubated for 15 min at room temperature. Last, the coverslip was gently washed three times with 200 µl of cytoskeleton buffer.

Cells were permeabilized by washing the coverslip once with 200 µl of permeabilization buffer (PEM [100 mM PIPES, pH 6.95; 2 mM EGTA; 1 mM MgCl$_2$] with 1% [wt/vol] bovine serum albumin (BSA)-fraction V and 0.3% [vol/vol] Triton X-100) and then incubated for 60 min upon a second addition of permeabilization buffer. Afterwards, 200 µl of the permeabilization buffer was replaced with primary antibodies diluted in permeabilization buffer, 1 µg/ml mouse DM1A anti-$\alpha$-tubulin antibody (Thermo Fisher Scientific; Cat. No. 62204) and 1:200 rabbit anti-Rosetteless (*Levin et al., 2014*). After the samples were incubated in primary antibody for 2 hr, the chamber was gently washed three times with 200 µl permeabilization buffer. Next, 200 µl of permeabilization buffer with 10 µg/ml donkey anti-mouse immunoglobulin G–AlexaFluor568 (Thermo Fisher Scientific; Cat. No. A10037), donkey anti-rabbit immunoglobulin G–AlexaFluor647 (Thermo Fisher Scientific; Cat. No. A32795), 10 µg/ml Hoechst 33342 (Thermo Fisher Scientific; Cat. No. H3570), and 4 U/ml Phalloidin-AlexaFluor488 (Thermo Fisher Scientific; Cat. No. A12379) was added to the chamber and then incubated for 40 min. Afterwards, the chamber was washed five times with PEM.

Immunostained samples were imaged on a Zeiss Axio Observer LSM 880 with an Fast Airyscan detector and a 40x/NA1.1 Plan-Apochromatic water immersion objective (Zeiss) by frame scanning in the superresolution mode with the following settings: 50 × 50 nm pixel size; 220 nm z-step; 0.73 µs/pixel dwell time; 750 gain; 488/561/633 nm multiple beam splitter; 633 nm laser operating at 16% power with a 570-620/645 bandpass/longpass filter; 561 nm laser operating at 16% power with a 570-620/645 bandpass/longpass filter; 488 nm laser operating at 14% power with a 420-580/495-550 bandpass filters.

## Next-generation sequencing (*Figure 4* and *Figure 4—figure supplement 1*)

We performed deep sequencing of edited cells to quantify the efficiency of genome editing (*Figure 3* and *Figure 4—figure supplement 1*). The transfections were performed as above with the following modifications: Two transfections were conducted for each condition and combined into 1 ml of low nutrient media (see *Supplementary file 1*-Table A for recipe). One day after transfection, the cells were harvested and dissolved in 50 µl of DNAzol direct (Molecular Research Center, Inc). Three independent transfections performed on different days provided replicate measures for each condition (*Figure 3C* and *Figure 4—figure supplement 1B*).

To preserve the diversity of sequences during PCR, six parallel PCR reactions (Q5 DNA polymerase, NEB) were set up with 30 µl of sample. The target locus was amplified in 15 thermal cycles, purified using magnetic beads (UC Berkeley DNA sequencing facility), and pooled together in a total volume of 180 µl. Importantly, the primers for this first round of PCR had a randomized 6-nucleotide sequence in the forward primer to distinguish PCR duplicates (primer sequences in *Supplementary file 1*-Table D), which allowed us to identify 4096 unique sequences. Extending this randomized sequence would result in higher sensitivity.

A second round of PCR was performed to attach adapters for Illumina sequencing. For these reactions, four replicate PCR reactions were set up with 25 µl of the purified products from the first round of PCR and primers with sequencing adapters and unique sample barcodes were attached in five thermal cycles. Afterward, the PCR products were purified using magnetic beads (UC Berkeley DNA sequencing facility) and their quality was assessed on a Bioanalyzer (UC Berkeley Functional Genomic Laboratory). The bioanalyzer traces showed that the amplicons were the proper size, yet a similar concentration of residual PCR primers remained in each sample. After quantifying DNA (Qubit; Thermo Fisher Scientific) and pooling equimolar amounts of sample, the amplicons were further purified with magnetic beads (UC Berkeley Functional Genomics Lab) and the concentration was verified using qPCR. The library was sequenced on a miSeq sequencer (Illumina, San Diego, CA) using the V3 chemistry (Illumina) for 300 rounds of paired-end sequencing, which gives up to 600 bases of sequence per sample. After sequencing the samples were separated based on their unique barcodes for further analysis of individual samples.

The editing efficiency for each sample was calculated from high-quality, unique reads. First, we used tools from the Galaxy Project (*Afgan et al., 2018*) to join paired-end reads into one read (fastq-join) and then retain high quality sequences (Galaxy–Filter by quality: 100% of bases with quality scores $\geq$ 30) with 50 bp of expected sequence from the *rosetteless* locus on the 5' and 3' ends of the amplicon (Galaxy–Cutadapt: 50 base overlap with 0.1 maximum error rate). Next, the reads were filtered for unique instances of the randomized barcode sequence from the first round PCR primers (Galaxy–Unique). We then combined matching amplicon sequences into unique bins, while counting the number of sequences in each bin (Galaxy–Collapse). The FASTA file of aligned sequences (Galaxy–ClustalW) from this initial processing was further analyzed using a custom script (*Source code 1*). To quantify the instances of template-mediated repair, we counted the number of sequences that had the PTS. Untemplated mutations were counted from insertions and deletions larger than 1 bp. The remaining sequences, those that were the same length as the *rosetteless* locus but did not have the exact amplicon sequence, were counted as single-nucleotide polymorphisms (SNPs). The outputs from each category were also visually inspected to reclassify incorrect calls, such as a few instances of template-directed repair in the insertions and deletion category due to mutations in the PTS. The SNP data revealed that conditions with or without the addition of *Sp*Cas9 or repair templates had same SNP frequency. Therefore, we only compared reads categorized as template-mediated repair or untemplated insertions and deletions. Importantly, this analysis may overlook some instances where DNA repair resulted in sequences that maintained the original sequence or introduced SNPs, thereby underestimating the efficiency of non-templated repair.

## Statistical analyses

Differences between experimental conditions were evaluated by comparing the variance between experimental groups, each with triplicate and biologically independent measures of the response variable, with Single- or Multi-Factor Analysis of Variance (ANOVA) (*Zar, 1999*). If the groups were very unlikely to have the same response ($p < 0.05$), the source of variation from Multi-Factor ANOVA was identified by performing Single-Factor ANOVA on subsets of the original experimental groups. Finally, a Tukey Multiple Comparison Test was performed after Single-Factor ANOVA showed that experimental groups were unlikely to have the same response. The specific analyses for each experiment are detailed in the following text:

*Figure 2*: A Two-Factor ANOVA was performed to test the effect of genotype on cell density in a serial dilution of cycloheximide, which was measured in three biological replicates. This test showed that the growth of the *rpl36a^{P56Q}* strain in cycloheximide was very unlikely to be the same as the wild-type strain ($p < 10^{-20}$).

*Figure 4*: Four sets of Two-Factor ANOVA tested the effect of repair template and *Sp*Cas9 cleavage on the percentage of PTS or INDEL mutations, as measured by deep sequencing of the *rosetteless* surrounding the position supercontig 8: 429,354 nt in three biological replicates of each condition. In the first and second sets of Two-Factor ANOVA, the outcomes of PTS mutations were separately analyzed from INDEL mutations, and within each set, we tested the effect of adding *Sp*Cas9 and repair templates on the mutation frequency, which showed that the frequencies of PTS or INDEL mutations were very unlikely to be the same in the presence of *Sp*Cas9 versus its absence ($p < 10^{-15}$ for PTS mutations and $p < 0.02$ for INDEL mutations). In the third and fourth sets of Two-Factor ANOVA, the editing outcomes in the absence of *Sp*Cas9 were separately analyzed from the

outcomes in the presence of *Sp*Cas9 as we tested the effects of mutation type (PTS versus INDEL) and the addition of repair templates on the mutation frequency. Importantly, the editing outcomes in the absence of *Sp*Cas9 were the same across all conditions (p<0.34), indicating that there is virtually no mutagenesis in the absence of *Sp*Cas9. Furthermore, in the presence of *Sp*Cas9, the editing outcomes across all repair templates were very improbably the same (p<$10^{-15}$). A follow-up, Single-Factor ANOVA that tested the effect of adding repair templates on the frequency of INDEL mutations in the presence of *Sp*Cas9 showed no differences between repair template groups (p<0.60). Whereas, the same analysis for PTS mutations in the presence of *Sp*Cas9 showed that editing frequencies were very unlikely to be the same (p<$10^{-4}$) across different repair templates. Therefore, multiple comparisons with the Tukey method were performed on all of the pairwise combinations of the repair template groups in the presence of *Sp*Cas9. These multiple comparisons showed that none of the repair templates resulted in the same frequency of PTS mutations. For example, the comparison that was most likely to have the same repair outcome was the single-stranded, sense oligonucleotide versus the duplex repair template (p<0.04); whereas, all other comparisons between repair templates were much less likely to be the same (0.0001 < p < 0.004). Among these repair templates, the single-stranded, sense oligonucleotide resulted in the highest average frequency of PTS mutations (1.57%).

*Figure 3—figure supplement 1*: After finding that both genotype and RIF induction affected the percentage of cells in rosettes with a Two-Factor ANOVA, in which three biological replicates were used for each group, a Single-Factor ANOVA followed by a Tukey Multiple Comparison Test was performed on groups that had been induced with RIFs to test the effect of genotype on rosette development. These tests showed that wild-type and *rpl36$^{P56Q}$* strains were similarly induced (p<0.84), which was also the case for *rtls$^{PTS1}$ rpl36$^{P56Q}$* and *rtls$^{tl1}$* (p<0.99), and rosette development was very unlikely to be the same between wild-type or *rpl36$^{P56Q}$* strains versus *rtls$^{PTS1}$ rpl36$^{P56Q}$* or *rtls$^{tl1}$* strains (p<$10^{-10}$). Note that such tests were not performed for the uninduced samples because the fraction of cells in rosettes was zero for all three measurements of each genotype.

*Figure 3—figure supplement 2*: The effect of genotype on growth rate was tested with a SIngle-Factor ANOVA using three biological replicate measures of the doubling time ($T$) for each strain, which showed that the doubling-times were probably the same among all strains (p<0.08).

*Figure 4—figure supplement 1*: The outcomes of PTS mutations were separately analyzed from INDEL mutations in two sets of Single-Factor ANOVA to test the effect of repair template in the presence of SpCas9 on mutation frequency. The frequency of INDEL mutations was most likely the same between all repair templates (p<0.17). In contrast, the frequency of PTS mutations was improbably the same between repair templates (p<$10^{-6}$), yet multiple comparisons with the Tukey test showed that there was no difference between the duplex templates (0.50 < p < 0.99) just as there were no differences between the single-stranded templates with truncated homology arms (0.99 < p < 1.00). Thus, the major effect on PTS mutation frequency was between the duplex templates versus the single-stranded, truncated templates (p<0.02).

## Acknowledgements

We thank the following people for insights and support that helped advance this work: Heather Szmidt-Middleton, Laura Wetzel, Monika Sigg, Lily Helfrich, Arielle Woznica, Sabrina Sun, Tara DeBoer, Jorge Santiago-Ortiz, and Kayley Hake helped with and provided feedback on early experiments. Through the Gordon and Betty Moore Foundation Marine Microbiology Initiative (GBMF MMI), Manny Ares first brought cycloheximide resistance alleles to our attention. The following people stimulated helpful discussions: Fyodor Urnov, Stephen Floor, James Gagnon, Chris Richardson, Jacob Corn, David Schaffer, Niren Murthy, Craig Miller, and members of the King Lab. Brett Stahl, Shana McDevitt, the UC Berkeley Vincent Coates Sequencing Center, and the UC Berkeley Functional Genomics Laboratory provided help with sequencing. This work was supported, in part, by a GBMF MMI Experimental Model Systems grant. DSB was supported through a Simons Foundation Fellowship from the Jane Coffin Childs Memorial Fund for Medical Research.

## Additional information

### Funding

| Funder | Grant reference number | Author |
|---|---|---|
| Gordon and Betty Moore Foundation | MMI Experimental Model Systems grant | David S Booth Nicole King |
| Howard Hughes Medical Institute | | Nicole King |
| Jane Coffin Childs Memorial Fund for Medical Research | Simons Foundation Fellowship | David S Booth |

The funders had no role in study design, data collection and interpretation, or the decision to submit the work for publication.

### Author contributions

David S Booth, Conceptualization, Resources, Data curation, Formal analysis, Funding acquisition, Validation, Investigation, Visualization, Methodology, Writing - original draft, Project administration, Writing - review and editing; Nicole King, Conceptualization, Supervision, Writing - original draft, Project administration, Writing - review and editing

### Author ORCIDs

David S Booth (iD) https://orcid.org/0000-0002-4724-4702
Nicole King (iD) http://orcid.org/0000-0002-6409-1111

### Decision letter and Author response

Decision letter https://doi.org/10.7554/eLife.56193.sa1
Author response https://doi.org/10.7554/eLife.56193.sa2

## Additional files

### Supplementary files

• Source code 1. Quantification of DNA repair outcomes. BASH script for quantifying the frequency of repair outcomes from deep sequencing data that were preprocessed and aligned in a Galaxy server (*Afgan et al., 2018*).

• Supplementary file 1. Tables of critical resources. **Table A:** Media recipes for making artificial seawater (*Hallegraeff et al., 2004*; *Skelton et al., 2009*), high nutrient media (modified from *King et al., 2009*; *Levin and King, 2013*; *Booth et al., 2018*), and low nutrient media. **Table B:** Oligonucleotide sequences for gRNAs, repair oligonucleotides, and primers that were used to construct and to validate genome edited strains. **Table C:** *S. rosetta* strains Genotypes and sources of *S. rosetta* strains used in this study. **Table D:** Deep sequencing library primers Sequences for primers (adapted from *Lin et al., 2014* used to generate libraries for deep sequencing (*Figure 4* and S5)

• Transparent reporting form

### Data availability

All data generated are included in the manuscript. Additionally, we have posted a protocol at https://www.protocols.io: https://doi.org/10.17504/protocols.io.89fhz3n.

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
