## [Decision Letter]

Thank you for submitting your article "Genome editing enables reverse genetics of multicellular development in the choanoflagellate *Salpingoeca rosetta*" for consideration by *eLife*. Your article has been reviewed by three peer reviewers, and the evaluation has been overseen by Alejandro Sánchez Alvarado as the Reviewing Editor and Patricia Wittkopp as the Senior Editor. The following individuals involved in review of your submission have agreed to reveal their identity: Margaret A Titus (Reviewer #1); Iñaki Ruiz-Trillo (Reviewer #2); Matthew C Gibson (Reviewer #3).

The reviewers have discussed the reviews with one another and the Reviewing Editor has drafted this decision to help you prepare a revised submission.

Summary:

This work represents an important step forward for the study of choanoflagellates. It reports the first evidence of a reverse genetics approach based on CRISPR/Cas-9 mediated genome editing in *Salpingoeca rosetta*, an emerging model to study the origin and evolution of animal multicellularity. In this work, authors engineered a cycloheximide-resistance selectable marker to demonstrate genome editing and to enrich for edited cells. Later they disrupted *rosetteless*, an *S. rosetta* C-type lectin gene essential for rosette colonies development, the absence of which impairs the formation of rosette colonies. This work represents an important advance to investigate in vivo the function of genes using a genome editing-based approach in *S. rosetta*, which will greatly expand its value as an experimental model to address animal origins.

Essential revisions:

1) The authors point out that the method here could also be used to edit the genome of related choanoflagellates. This would assume that the transfection method would work as well on these (or some of them) as well as it does for *S. rosetta*. Has this been tested or is there a strong reason to think that this will be the case, especially for the more divergent members of the species?

2) One desired use of gene editing would be to introduce a tag to a gene of interest for expression of an in-frame fusion at endogenous levels. The insertions made are rather small in size. Have the authors tried to fuse a GFP, for example, to a gene of interest and found this to work? Or do the data the authors have in hand support the possibility to insert a larger DNA to generate such a fusion?

3) One consideration with gene editing is the introduction of off-target mutations. Do the authors have any information about the possibility of generating additional, background mutations in the genome when using the co-CRISPR approach?

4) One nagging question in CRISPR/Cas9 reverse genetic approaches is the possibility of off-target mutagenesis, which is most effectively ruled out by 1)independent gRNAs; 2) evidence for Mendelian segregation and 3) non-complementation between independent alleles in a diploid organism. All experiments shown here are performed in haploid cells. However, *Salpingoeca rosetta* can undergo sexual reproduction through a diploid state (Levin and King, 2013) with genetic analysis suggesting that inheritance patterns follow Mendel's law of segregation and independent assortment (Levin et al., 2014). Can CRISPR/Cas9 mutagenized choanoflagellates undergo sexual reproduction? If so, is allele frequency in line with the rules of classical Mendelian genetics? How could this be utilized in future experiments to improve genetic resolution? A discussion of this information, if possible, would help to fully expand on the potential uses for targeted mutagenesis in *S. rosetta*.

5) In the manuscript version for review, the protocol availability on protocol.io appears only once – towards the middle of the Materials and methods (subsection “Delivery of gene editing cargoes with nucleofection”). I would suggest this information additionally be listed in the Abstract and/or somewhere in the main body of the paper.

6) By directly showing the loss of *Rosettless* protein in the mutants, Supplementary Figure 3C offers striking visual corroboration of the genetic and genomic evidence of targeted mutagenesis. I found this to be particularly strong data and would suggest moving these panels to the main text. Within the figures, it would also be helpful to readers to clearly indicate the color key for the markers used somewhere in the figure itself (*Rosetteless* (green), α-tubulin (gray), and actin (magenta).

7) The main Figure 1 is essentially review information and the main Figure 2 is important but graphically and conceptually quite simple. The authors should consider fusing these figures by putting Figure 1 in a horizontal format with the Figure 2 panels below. I think this would be more engaging for readers and also open up room to move Supplementary Figure 3 wholesale into the main presentation.

8) The authors provide detailed analysis to show that the inclusion of a donor template significantly favors DNA repair over indel creation (Figure 4, Figure 4—figure supplement 1). However, there appears to be no significant difference in editing outcomes with different types of repair templates as long as the template has both left and right homology arms. Maybe I'm missing something, but this seems at odds with the final sentence of the Results section.

---

## [Author Response]

Essential revisions:1) The authors point out that the method here could also be used to edit the genome of related choanoflagellates. This would assume that the transfection method would work as well on these (or some of them) as well as it does for *S. rosetta*. Has this been tested or is there a strong reason to think that this will be the case, especially for the more divergent members of the species?

This point highlights that a limiting step for performing genome editing and reverse genetics in *S. rosetta* and other choanoflagellates has been the establishment of an efficient method for transfection. In our previous paper that focused on transfecting *S. rosetta* (Booth et al., 2018), we described how transfection could be applied to other systems. In this paper, we raise the possibility of establishing genome editing in other choanoflagellates (Discussion) as a hypothetical (“…we anticipate…”) based on lessons that we’ve learned from *S. rosetta*.

In fact, we have helped advise another group that has adapted the transfection procedure from *S. rosetta* to successfully transfect another choanoflagellate species. As these data are still unpublished and not ours to report, we think it is most appropriate to simply raise the possibility that our methods might extend to other choanoflagellates.

2) One desired use of gene editing would be to introduce a tag to a gene of interest for expression of an in-frame fusion at endogenous levels. The insertions made are rather small in size. Have the authors tried to fuse a GFP, for example, to a gene of interest and found this to work? Or do the data the authors have in hand support the possibility to insert a larger DNA to generate such a fusion?

We agree that one of the desired uses for gene editing is the insertion of fluorescent protein tags at endogenous loci. Although we do not currently have data that demonstrate the incorporation of such tags, we are actively pursuing these experiments (when we can return to lab, that is). It is worth noting that even in large research communities, such as researchers using human cell lines and/or established model organisms, there has often been a long wait between establishing genome editing and inserting large tags (Hsu et al., 2014) with more and more clever insights continually being published (e.g. Li et al., 2019). Nonetheless, with the tools in hand, we are optimistic that small epitope tags could be incorporated at endogenous loci to accelerate protein localization experiments. As we have not yet been able to pursue those experiments, we have not commented on that particular use in the submitted manuscript.

3) One consideration with gene editing is the introduction of off-target mutations. Do the authors have any information about the possibility of generating additional, background mutations in the genome when using the co-CRISPR approach?

We agree that an important concern for any CRISPR-based approach is the possibility of introducing off-target mutations. In fact, this potential problem was a key motivation behind our decision to perform genome editing with the *Sp*Cas9 RNP (as opposed to a *cas9* plasmid), as previous studies comparing off-target mutations with different Cas9-delivery methods showed fewer off-target mutations with the Cas9 RNP (Kim et al., 2014; Liang et al., 2015; Han et al., 2020). We have now added this point and associated references to our list of reasons for using the *Sp*Cas9 RNP:

“We favored delivering the *Sp*Cas9 RNP rather than expressing *Sp*Cas9 and gRNAs from plasmids, as RNA polymerase III promoters for driving gRNA expression have not yet been characterized in *S. rosetta* and the overexpression of *Sp*Cas9 from plasmids can be cytotoxic for other organisms (Jacobs et al., 2014; Jiang et al., 2014; Shin et al., 2016; Foster et al., 2018) as well as increase the likelihood of introducing off-target mutations (Kim et al., 2014; Liang et al., 2015; Han et al., 2020).”

Although we did not directly measure the frequency of off-target mutagenesis during genome editing in *S. rosetta*, genetic crosses have been performed in *S. rosetta* (Levin et al., 2014; Wetzel et al., 2018) and offer a means to eliminate background mutations for future studies (also see our response to comment 4). Moreover, circumstantial evidence suggests that off-target mutagenesis may be infrequent in *S. rosetta.* Off-target mutagenesis in other organisms typically occurs through untemplated insertions and/or deletions (Kim et al., 2014; Liang et al., 2015; Han et al., 2020), but these are rare according to our next generation sequencing at the *rosetteless* locus (Figure 4).

4) One nagging question in CRISPR/Cas9 reverse genetic approaches is the possibility of off-target mutagenesis, which is most effectively ruled out by 1)independent gRNAs; 2) evidence for Mendelian segregation and 3) non-complementation between independent alleles in a diploid organism. All experiments shown here are performed in haploid cells. However, Salpingoeca rosetta can undergo sexual reproduction through a diploid state (Levin and King, 2013) with genetic analysis suggesting that inheritance patterns follow Mendel's law of segregation and independent assortment (Levin et al., 2014). Can CRISPR/Cas9 mutagenized choanoflagellates undergo sexual reproduction? If so, is allele frequency in line with the rules of classical Mendelian genetics? How could this be utilized in future experiments to improve genetic resolution? A discussion of this information, if possible, would help to fully expand on the potential uses for targeted mutagenesis in *S. rosetta*.

Thank you for encouraging us to include these points for discussion. Based on this comment, we have added the following paragraph to the Discussion to better highlight how independent alleles can work together to establish genotype to phenotype relationships, just as we show in this manuscript with two *rosetteless* alleles (Figure 3). We also highlight how genetic crosses and transgenesis could further aid reverse genetics.

“Moving forward, the approach established here promises to accelerate future research on choanoflagellates. […] Tools that have been previously used in forward genetic approaches, such as genetic crosses (Levin and King, 2013; Levin et al., 2014; Woznica et al., 2017) and stable transgenesis (Wetzel et al., 2018), may also provide the means to rapidly generate strains with different genetic backgrounds to complement mutants, to reveal epistasis, or simply to eliminate off-target mutations that may arise during genome editing.”

To respond to the specific question about whether genome edited strains can mate, we do expect that genome edited strains can undergo sexual reproduction, and we have some preliminary evidence, but we feel this goes beyond the scope of the current study and the relevant data do not yet meet our standards for publication. As previous forward genetic studies have shown, the frequency of alleles in haploid progeny from genetic crosses do obey classical Mendelian genetics. Except for tightly linked loci and potential epistasis between alleles, we would expect that these results should be generalizable to genome edited strains.

5) In the manuscript version for review, the protocol availability on protocol.io appears only once – towards the middle of the Materials and methods (subsection “Delivery of gene editing cargoes with nucleofection”). I would suggest this information additionally be listed in the Abstract and/or somewhere in the main body of the paper.

Thank you for this suggestion. We have added a mention of the protocol in the Introduction:

“Here we report a reliable method for genome editing to perform reverse genetics in *S. rosetta* that we have developed into a publicly-accessible protocol (https://dx.doi.org/10.17504/protocols.io.89fhz3n).”

6) By directly showing the loss of Rosettless protein in the mutants, Supplementary Figure 3C offers striking visual corroboration of the genetic and genomic evidence of targeted mutagenesis. I found this to be particularly strong data and would suggest moving these panels to the main text. Within the figures, it would also be helpful to readers to clearly indicate the color key for the markers used somewhere in the figure itself ( Rosetteless (green), α-tubulin (gray), and actin (magenta).

Done.

7) The main Figure 1 is essentially review information and the main Figure 2 is important but graphically and conceptually quite simple. The authors should consider fusing these figures by putting Figure 1 in a horizontal format with the Figure 2 panels below. I think this would be more engaging for readers and also open up room to move Supplementary Figure 3 wholesale into the main presentation.

Thank you for these suggestions. Following these recommendations, we have moved the immunofluorescence images from the supplement to the main text Figure 3G-J. Although simple, we think Figures 1 and 2 make distinct points for readers to take away and would like to keep them as separate figures.

8) The authors provide detailed analysis to show that the inclusion of a donor template significantly favors DNA repair over indel creation (Figure 4, Figure 4—figure supplement 1). However, there appears to be no significant difference in editing outcomes with different types of repair templates as long as the template has both left and right homology arms. Maybe I'm missing something, but this seems at odds with the final sentence of the Results section.

Thank you for this question, as it emphasizes that we need to clarify our data regarding relative rates of repair with different templates. Although the frequencies of genome editing appear largely comparable among the repair templates (Figure 4C; subsection “*S. rosetta* preferentially introduces genome-edited mutations from DNA templates”), there are meaningful differences that may be relevant for experimental design. In Figure 4C, it may have been difficult to discern those differences because the logarithmic y-axis compresses values and deemphasizes the variance and differences between experiments with different repair templates, so we have revised this panel to extend the y-axis. We have also underscored the differences between each repair template by reporting in the text the P-values from ANOVA and multiple comparison tests. To the reviewers’ point, we also emphasize in the main text that the editing frequencies are similar for all practical purposes.

“We found that the *S. rosetta* genome could be edited in a *Sp*Cas9-dependent manner using a variety of templates (Figure 4C, Figure 4—figure supplement 1C). In the presence of the *Sp*Cas9, INDEL mutations occurred at a frequency of < 0.1%, either in the presence or absence of DNA repair templates (Figure 4C, Figure 4—figure supplement 1D-E). In contrast, the addition of *Sp*Cas9 with DNA repair templates that encoded the PTS resulted in PTS mutations at a frequency of 0.79-1.57%, which is at a ten-fold higher frequency than the INDEL mutational frequency (Two-Factor ANOVA: *P* < 10^-13^). Notably, the total frequency of genome editing (~ 1%) is on the same order of magnitude as transfection efficiency (~1%; Booth et al., 2018), suggesting that the delivery of *Sp*Cas9 and repair templates is the biggest factor limiting genome editing efficiency.

In the absence of *Sp*Cas9, we observed two types of *Sp*Cas9-independent genome edits. The first was a single INDEL mutation detected in a population of cells transfected with the antisense repair template, and the second was the detection of PTS mutations at an average frequency of ~ 0.02% upon the delivery of a double-stranded repair template. Although the frequency of these mutations occurred at a rate less than or equal to the detection threshold (~ 0.02%), meaning that we could not confidently conclude that differences exist between any of the samples (Two-Factor, ANOVA: *P* < 0.27), the presence of these mutations is consistent with a low frequency of endogenous DNA repair. These results also emphasize that the addition of *Sp*Cas9 was essential for efficient, targeted mutagenesis (Two-Factor ANOVA: *P* < 0.02 for INDEL mutations and *P* < 10^-13^ for PTS mutations).

Altogether, our optimization efforts revealed that the delivery of the *Sp*Cas9 with a DNA template spanning both sides of the *Sp*Cas9 cleavage site introduced PTS mutations at a frequency of ~ 1%. We recommend using a sense-oriented, single-stranded template for genome editing, as this template led to the highest frequency of PTS mutations (Single-Factor ANOVA, Tukey Multiple Comparison Test: *P < 0.04*) and costs less to synthesize than a double-stranded template.”

We also point out other considerations that have led us to favor the sense repair template. First, single-stranded oligonucleotides for the sizes of repair templates that were used in this paper are cheaper and faster to obtain compared to duplex templates. Second, the single-stranded oligo in the sense direction leads to higher editing frequencies (Figure 4 and Figure 2—figure supplement 2E, compare templates with 20-30 base homology arms). Given these collective considerations, we argue that the single-stranded, sense template represents the all-around best guide template to be used by other researchers who wish to implement these methods for their own research.